# Making information and communications technologies (ICTs) work for health: protocol for a mixed-methods study exploring processes for institutionalising geo-referenced health information systems to strengthen maternal neonatal and child health (MNCH) service planning, referral and oversight in urban Bangladesh

Rubana Islam [1] Alayne Mary Adams [2] Shaikh Mehdi Hasan [3] Rushdia Ahmed [3] Dipika Shankar Bhattacharyya [3] Sohana Shafique [3]

For numbered affiliations see end of article.

**Correspondence to**
Dr Sohana Shafique;
sohana.shafique@icddrb.org

## ABSTRACT

**Introduction** Disparities in health outcomes and access to maternal neonatal and child health (MNCH) are apparent among urban poor compared with national, rural or urban averages. A fundamental first step in addressing inequities in MNCH services is knowing what services exist in urban areas, where these are located, who provides them and who uses them. This study aims to institutionalise the Urban Health Atlas (UHA)—a novel information and communications technology (ICT) tool—to strengthen health service delivery and oversight and generate critical evidence to inform health policy and planning in urban Bangladesh.

**Methods and analysis** This mixed-method implementation research will be conducted in four purposively selected urban sites representing larger and smaller cities. Research activities will include an assessment of information needs and task review analysis of information users, stakeholder mapping and cost estimation. To document stakeholder perceptions and experiences, key informant interviews and in-depth interviews will be conducted along with desk reviews to understand MNCH planning and referral decisions. The UHA will be refined to increase responsiveness to user needs and capacities, and hands-on training will be provided to health managers. Cost estimation will be conducted to assess the financial implications of UHA uptake and scale-up. Systematic documentation of the implementation process will be undertaken. Policy decision-making and ICT health policy process flowcharts will be prepared using desk reviews and qualitative interviews. Thematic analysis of qualitative data will involve both emergent and a priori coding guided by WHO PATH toolkit and Policy Engagement Framework.

## Strengths and limitations of this study

► This mixed-method implementation research is among the first in Bangladesh to explore processes for institutionalising geo-referenced health information systems to strengthen maternal neonatal and child health service planning, referral and oversight in urban areas.

► A conceptual framework specific to information and communications technology (ICT) tools and their implementation has been developed to guide the research.

► Implementation partnership with the government will be established to ensure post-implementation maintenance of the geo-referenced health information system.

► Acknowledging the potential contribution of patients and civil society groups to ICT uptake, these groups are not engaged in the health planning exercise for logistic reasons.

Stakeholder analysis will apply standard techniques and measurement scales. Descriptive analysis of quantitative data and cost estimation analysis will also be performed.
**Ethics and dissemination** The study has been approved by the Institutional Review Board of icddr,b (# PR-16057). Study findings will be disseminated through national and international workshops, conferences, policy briefs and peer-reviewed publications.

## INTRODUCTION

Bangladesh has embraced the Sustainable Development Goal-3 (SDGs) of achieving



universal health coverage by 2030[1]; however, challenges related to rapid population growth, pluralistic health systems and lack of governance, among others, are substantial.[2] Although Bangladesh has made extraordinary progress in reducing maternal and child mortality,[3] there are significant disparities in health-related outcomes and access to maternal neonatal and child health (MNCH) services stratified along both socioeconomic and geographic dimensions. Health indicators are far worse in urban slums than both non-slum urban areas and the national average.[4] Nationally, the mortality rate for children under 5 years of age is 65 per 1000 live births and 49 per 1000 live births in rural areas, while the rate is 81 per 1000 live births among urban slum residents.[4 5] Undergoing rapid urbanisation, the country is projected to become over 50% urban by 2040, with almost one-third of urban residents living in slums.[6] Persistent inequities in key MNCH indicators in urban areas highlight the need to focus on issues of service coverage, access, quality, and timely and appropriate referral as urgent policy priorities.

The Bangladesh urban health system is a smorgasbord of service providers characterised by inadequate coordination and regulation, and geographic and socioeconomic inequities in healthcare access.[7 8] Several reasons have been proposed for inefficiencies in the system including poor planning and management capacity, weak coordination among the authorities, lack of clear, separate roles and responsibilities for the various authorities, service coverage gaps and human resource management issues.[9 10] Of particular concern in urban areas is the lack of adequate public primary care infrastructure and services which disproportionately impacts the urban poor, and poses significant challenges to the country's aspirations to meet the goal of Universal Health Coverage by 2030.[11] One consequence of limited formal primary healthcare services in urban areas[12] is the emergence of the private sector in health including the proliferation of informal providers such as pharmacies on which many of the urban poor rely. The formal private sector is equally massive, accounting for 80% of over 3500 hospitals in Bangladesh, and an even greater percentage in urban areas.[13] Lack of regulation of this sector has resulted in concerns about quality of care and financial accessibility, especially for the urban poor.[10]

A fundamental first step in addressing inequities in urban healthcare access is an in-depth understanding of what services exist, their location, who provides them and who uses them. A strong health management information system (HMIS), an essential component of sound programme development and implementation. Enabling the use of data for strategic decision-making, better governance, institutionalised HMIS systems represent the foundation on which improvements in health outcomes can be monitored and greater accountability ensured.[14 15] A master facility list (MFL) is a crucial constituent of HIS and permits the linkage of subsystems within national HIS architecture.[16 17] MFL is advocated by the WHO as an effective means of ensuring better governance including systematic reporting and monitoring supervision.[18 19] MFLs like Urban Health Atlas (UHA) are expected to facilitate health service planning and management through mapping or visualising the distribution of health services and resources. It can also assist health service providers in identifying appropriate referral facilities for patients.[20] These functions can help improve equitable service coverage and reduce delays in receiving appropriate care, which in turn can impact health outcomes such as maternal and child mortality among the urban poor. A theory of change is provided in online supplemental file 1.

Realising the critical role that health information systems play in health management and building on political commitment towards 'Digital Bangladesh', the Bangladesh Directorate General of Health Services (DGHS), Ministry of Health and Family Welfare (MOHFW) is implementing the District Health Information Software-2 (DHIS2) with support from development partners. While the system has been rolled out nationally, information is largely confined to public healthcare facilities. In urban areas, with the exception of large public hospitals and a number of non-government organisations (NGOs) involved in primary care provision, data are particularly sparse, especially for the massive private sector.

## Urban Health Atlas: a novel ICT tool

Addressing this information gap, icddr,b has created a geo-referenced heath facility database for nine major cities and municipalities across Bangladesh. This dataset consists of a census of all healthcare facilities and the services they provide along with their geolocations.[21] To enable the practical application of this dataset, an information and communications technology (ICT) tool called *Urban Health Atlas* (UHA) was developed (http://urban-healthatlas.com).[22] This GIS-based interactive online tool displays health facility data visually and permits their manipulation for better healthcare planning and decision-making. Providing detailed information on the location and services available at public and private health facilities, it allows users to examine gaps and duplication in service provision, assess the coverage of emergency services and the availability of doctors in a 24-hour period, calculate the shortest distance to referral facilities from any location and determine whether a given facility is licensed and registered. This information is particularly useful in helping healthcare planners and policy makers make informed decisions around the distribution and monitoring of healthcare facilities and services, and health human resources. For the general public, the tool holds promise in locating a desired healthcare service that is closest in distance and indicating the shortest path to get there.

A key strength of this dataset is its inclusion of private-for-profit healthcare facilities, from pharmacies to hospitals, in addition to public and private not-for-profit healthcare provision. The UHA prototype has been demonstrated both nationally and internationally, and

generated a great deal of interest and useful feedback. In Bangladesh, its promise has intrigued multiple stakeholder groups, ranging from the DGHS responsible for national healthcare planning, local government officials, private not-for-profit or (NGOs, service providers and development partners. In the context of significant investments in urban health systems strengthening that are in pipeline, and absence of urban data in the country's national health information system (DHIS2), UHA is widely regarded as timely and useful in the context of current urban health planning processes, and many discussions about its formal linkage to and institutionalisation within existing health information systems have occurred.

However, due to the complexity of these kinds of data, they risk being underutilised for health policy and planning unless specific efforts are attempted to make them more accessible to non-technical, policy and other local level stakeholders.[23] In the context of UHA, these efforts have included making the data available on the DGHS web page, and organising dissemination events in city corporations and municipalities. However, beyond anecdotal reports, there is no systematic information on whether the tool is being used by stakeholders, and how it could be improved to better meet their needs. The purpose of this study, therefore, is to pilot and refine the UHA for use in service delivery planning and referral, and by generating evidence on its utility, inform and strengthen advocacy for and action around its institutionalisation into the government system. A focus on MNCH service delivery was chosen to circumscribe the development of training materials and to clearly delimit the range of stakeholders that should be engaged.

## Study aims

Three specific aims are identified in seeking to institutionalise the UHA for MNCH service delivery, planning and referral into the government system:

1. To document stakeholder perceptions and experiences in adopting a tool that enables use of health facility information for strategic planning, day-to-day decision-making, control and oversight, and improved administrative efficiency of urban MNCH services.
2. To identify policy and programmatic entry points that will facilitate broader use of geo-referenced health facility information and its regular update.
3. To estimate costs associated with bringing geo-referenced facility listing into the government system.

## METHODS

This implementation research (IR) focuses on the factors and processes that influence uptake, use and scale-up of ICT tools like UHA. The study will explore barriers in usability, understandability and utility, as well as policy and other requirements needed to support its systematic implementation in the real-world setting of healthcare planning, referral and oversight. The primary audiences

of this research are managers and decision makers in the urban healthcare sector of Bangladesh.

## Study design and participants

The proposed implementation research employs a mixed-method research approach. Mixed-method research is a widely used approach in IR.[24] We will assess the uptake of UHA by the MNCH-related planners and decision makers over a 3-year period from 2016 to 2019. The specific IR variables to be assessed are adoption, appropriateness, feasibility and implementation cost.[24] Many IR frameworks exist, however, it is advised to use a framework befitting programme parameters.[24] For this reason, we identified a toolkit specific to the development and use of ICT tools and formulated our conceptual framework accordingly. This helped to operationalise the research as an ICT intervention versus a clinical or health service level intervention, while still retaining some of the features of common IR frameworks, including concerns with guided implementation and innovation, sustainability and stakeholder input.[24]

The WHO PATH toolkit was published by WHO and PATH to guide the introduction and implementation of ICT in health information systems.[19] The introduction of a new ICT tool is commonly accompanied by challenges that must be overcome. Before scaling-up, therefore, it is important to conduct rigorous product planning and feasibility testing, and to identify and engage key stakeholders. The toolkit identifies three main phases of an ICT project: pilot, scale and sustain. In the pilot phase, the phase addressed in this study, a solution (UHA) is developed based on programme needs and priorities; and tested on a small scale to measure outcomes, impacts and costs; and identify potential improvements. Several other factors influence the introduction of the HMIS in developing countries including planning, stakeholder roles and responsibilities, cultural aspects, human capacity, financial aspects sustainability (see figure 1).[25] The elements of this ICT-informed framework are similar to that of the different phases of the 'replicating effective programmes framework'used in IR.[26] For instance, stakeholder needs in our framework are addressed under the identifying implementation barriers step (pre-conditions phase), the orientation step of the pre-implementation phase is similar to participation and awareness and financial aspect elements in our framework, and training and technical assistance of the implementation phases are addressed through the human capacity development component.

In order to explore user perceptions, and policy and programmatic entry points, the Policy Engagement Framework[23] also will be employed to prospectively analyse policy that incorporates strategies for change. This framework will confer a systematic approach to the ongoing collection, analysis and use of political information (eg, concerning actors, their interests, institutions, ideas, and policy processes and context) that can alter the balance of power between those in support of and those

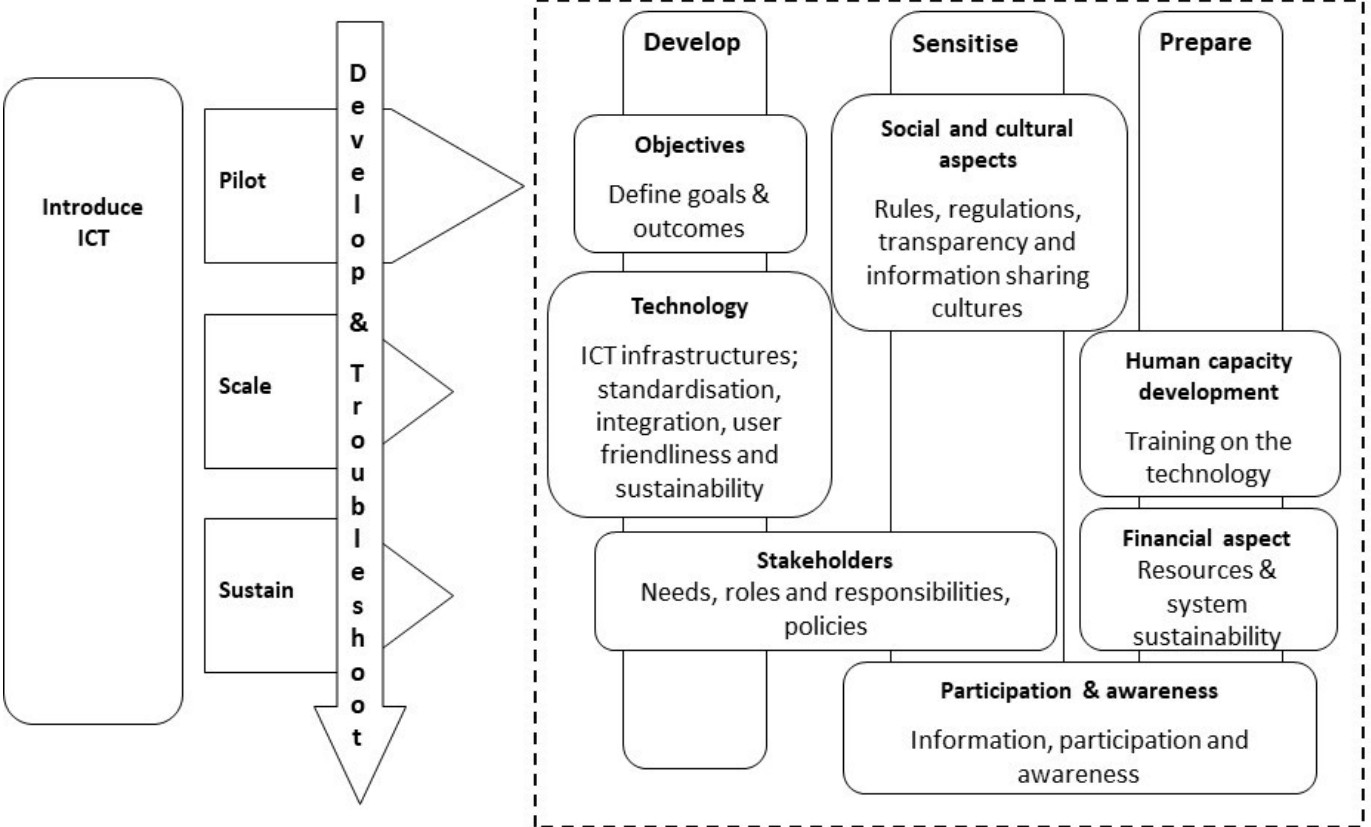

**Figure 1** Modified conceptual framework for information and communications technology (ICT) implementation in developing countries.

resisting change by enabling pro-reformers to intervene more effectively in the policy process.[23]

### Study sites
Two city corporations (CCs)—namely, Dhaka North City Corporation (DNCC) and Dhaka South City Corporation (DSCC)—and two municipalities, Jessore and Dinajpur, which have geo-referenced health facility data available from previous mapping exercise of icddr,b[22] will be purposively selected. As smaller cities, Jessore and Dinajpur municipalities present marked differences from CCs in terms of size, structure, capacity and challenges posed.

### Sample size
Through the stakeholder mapping, key urban MNCH decision-making actors at both national and local levels will be engaged to help identify potential users for UHA. This group will constitute our study respondents and training participants. When determining sample size for qualitative research, Guest *et al* propose that a homogeneous group of respondents' 12 interviews is sufficient for reaching data saturation.[27] It is also asserted that a sample under 20 respondents allows qualitative researchers to establish and maintain effective relationships with study participants, and thus enhances the validity of the research.[28] For these reasons, we will sample 15–16 respondents for each of our activities. The sampling

strategy, type and number of respondents for each study activity are provided in table 1.

### Implementation procedure of the pilot
The study implementation is envisaged to begin with stakeholder sensitisation and partnership building with government. This is critical because the research will work closely with the government health system in urban areas. In addition, an intervention will be conducted consisting of capacity building sessions around the use of the UHA for government and other urban health planners and managers. Details of these activities are provided below.

#### Stakeholder consultation and engagement
Two stakeholder consultation workshops will be carried out to identify and engage key stakeholders to create research buy-in and to begin the process of UHA advocacy. Detailed information on identification and mapping of stakeholders has been described in the Data collection section. Ideas will be generated for uptake, regular use and update of UHA.

#### Partnership development with government
Implementation partnerships with the Management Information System (MIS) of the DGHS, MOHFW will be developed and a Memorandum of Understanding will be signed between icddr,b and DGHS. Permission letters will also be obtained from mayors of CCs and municipalities.

**Table 1** Sampling strategy and sample type for each activity of the study

| Activity and focus | Data collection methods | Sampling strategy | Respondent group | Sample size |
|---|---|---|---|---|
| **Task review** (document how MNCH planning and referral decisions are currently made) | KIIs | Opportunistic/ emergent sampling; snowball sampling | ► Urban health systems actors<br>► National and local government officials<br>► NGO programme managers | 16 |
| | Desk review | N/A | N/A | N/A |
| **User need and user experience** (explore user preferences, task needs and experiences) | IDIs (on user needs) | Opportunistic/ emergent sampling; snowball sampling | ► Policy makers within the MOHFW<br>► Members of the Urban Health Cell of MOLGRDC<br>► Managers at City Corporations<br>► NGO programme managers | 16 |
| **Policy engagement** (understand interactions between content, context, actors and processes of policy advocacy and entry points for sustainable incorporation of ICT into health systems) | KIIs | Opportunistic/ emergent sampling; snowball sampling | ► Urban health systems actors<br>► National and local government officials<br>► NGO programme managers | 15 |
| | Desk review (urban health and ICT-related policies) | N/A | N/A | N/A |

ICT, information and communications technology; IDI, in-depth interview; KII, key informant interview; MNCH, maternal, newborn and child health; MOHFW, Ministry of Health and Family Welfare; MOLGRDC, Ministry of Local Government, Rural Development and Cooperatives.

### Development of training materials

A training manual on UHA will be prepared to guide UHA capacity building workshops including case studies, guidelines for group work and hand-on activities, pre-test/post-test questionnaires and so on.

### UHA workshops

In each study site, a 2-day UHA workshop, and subsequent 1-day refresher course, will be organised with a selected group of health workers and managers drawn from local government, and NGOs. Institutional agreements and permissions will be sought in advance from local government institutions and the health ministry as appropriate. Training sessions will provide an introduction to current urban health challenges, followed by an overview and demonstration of the UHA and its functions. Hands-on training, group work and case studies will be undertaken to familiarise users with UHA and to get their feedback on how it might be improved to better meet their needs.

### Data collection

Interviews will be conducted by an experienced group of researchers trained in qualitative interviewing including a mixed-method expert, two software programmers, one GIS expert and an economist. The team will begin data collection in Dhaka, then move to the municipalities. A period of rapport building with key stakeholders in each

study site will be critical to the success of this research given known difficulties in accessing the government sector. Using existing networks and negotiation skills will be especially important in opening doors and initiating discussion. The UHA tool will be assessed for impact on MNCH decision-making and outcomes. It is unlikely, however, that these effects would be apparent within 3 years of implementation. Thus, success of the tool will be determined based on user experiences as specified by WHO PATH Toolkit, that is, better indicators for strategic planning, day-to-day decision-making, control and oversight, and reduced administrative burden.

### Data collection methods for objective 1

To address objective 1, assessment of information needs and task review analysis will be done by desk review, in-depth interviews and click streams. Guidelines for qualitative interviews will be developed based on the WHO PATH toolkit's questions to measure success. Three qualitative research activities are envisaged to address this objective:

1. Key informant interviews (KIIs) with urban health system actors along with desk reviews to understand and document how MNCH planning and referral decisions are currently made.

2. In-depth interviews (IDIs) with potential UHA users to explore user preferences and task needs to refine the tool in advance of training.
3. IDIs with UHA users to understand their experiences and to document challenges and successes of using UHA for MNCH service decision-making during training and 1 and 3 months post-training.

In addition to qualitative assessments of user experiences, quantitative assessments of how different stakeholders are using data remotely will be made through (1) user's click streams; (2) task time devoted to different applications. Written feedback through online tools (ie, Google Analytics) that facilitate remote testing will also be collected to generate more user-friendly functions that meet user needs.

### Data collection methods for objective 2

To identify entry points that will facilitate broader use of geo-referenced facility information and its regular update, stakeholder mapping, policy mapping using desk review, KIIs and stakeholder consultation workshop will be conducted. Guidelines for qualitative interviews will be developed using the Policy Engagement Framework as a guide. Policy and programmatic entry points for the broader use and update of facility information, stakeholder analysis will be undertaken using Policy Engagement Framework as a guide. Stakeholders are identified as persons, groups, organisation members or systems that affect or can be affected by a project/programme/activity. Stakeholder analysis is an approach for generating knowledge about roles, behaviour, inter-relation and intention of associated actors and their influence in implementation processes of a programme or policy.[29] Given the importance of stakeholder satisfaction and support for the success of any programme,[30 31] incorporating stakeholders' perspectives and needs is a critical step in gaining ownership around an ICT innovation like UHA and its incorporation into routine information systems, and use for decision-making. The following qualitative methods will be used to fulfil objective 2:
1. Stakeholder mapping including the identification and listing of stakeholder groups involved in urban health based on available literature and expert opinion.
2. A semistructured guideline will be used to collect information during stakeholder consultation workshops to explore their respective interests, roles and responsibilities in urban health, their information needs and perceptions of how they can contribute to institutionalising UHA.
3. KIIs along with desk reviews will be undertaken to understand the processes of current health policy-making mechanisms and what other policies affect the integration of ICT in health.

### Data collection methods for objective 3

To estimate cost of bringing facility listing into the government information system, the total cost of ownership for UHA development and implementation will be estimated using an ingredient approach. Data will be collected through structured questionnaire, document review and KII. Cost will be estimated based on supply-side aspects. The budget matrix will be developed with cost drivers proposed in the WHO PATH toolkit.

The cost for development and implementation of UHA tools, coordination and engagement of city corporations and DGHS will be estimated including both financial costs and economic costs of the programme. Financial costs represent the actual expenditures on goods and services purchased. Economic costs include the estimated value of goods or services for which either there are no financial transactions or the price of a specific good did not reflect the cost of using it productively elsewhere.[32] The cost will be separated for start-up cost and implementation cost. The implementation cost comprises the costs required to run and maintain the ICT tools while executing intervention.[33]

All supply-side inputs will be identified, quantified and valued through a facility-level inventory, record reviews and KIIs. Both fixed cost and variable cost will be captured. Shared cost items, including salary, buildings, furniture, supervision, transportation and vehicles, will be identified through observation and interviews of relevant personnel. Shared costs will be apportioned by proportion of the time involvement of the relevant items (ie, office rent, common vehicle) to different activities. The time of volunteers will be transformed into costs by using the minimum wage level of manual workers in Bangladesh. There are some examples of components included in ingredient approaches (table 2)[33 34]:

To identify activities of the UHA tool implementation and their related inputs, a review of programme documents and interviews with relevant personnel will be conducted. The unit price/salary information will be collected from responsible programme management. In case of missing unit price of any items, the market price of those items will be collected. Semistructured checklists for cost data collection will be developed considering the programme context and using WHO PATH toolkit as a guiding framework. The budget matrix will be completed with the help from key personnel associated with costing and budgeting identified during the stakeholder mapping and research team's own estimates.

**Table 2** Components included in ingredient approaches for cost

| Method name | Methodology | Inputs | Activities |
|---|---|---|---|
| Ingredients approach | Quantities×price, personnel, percentage use | Personnel, ICT tools, vehicles | Personnel, ICT tools, vehicles |

.ICT, information and communications technology.

**Table 3** Operational definitions for stakeholder analysis for policy engagement

| Theme | Terms used | Operational definition |
|---|---|---|
| **Influence, importance and agreement analysis of stakeholders** | Level of influence | Stakeholders' influence will be determined according to each stakeholder group's perception and views on who is important in terms of urban healthcare delivery |
| | Level of agreement | Stakeholders' agreement will be determined according to how much each stakeholder agreed |
| | Level of importance | The stakeholders' importance will be determined according to how important each stakeholder group is to the other groups |
| **Power and leadership** | Overall power | Power of a stakeholder group will be assessed as compared with all other groups in Bangladeshi urban healthcare delivery system. Power of stakeholders will be measured as the product of multiplication of influence and importance. |
| | Relative position | Relative position of each stakeholder group will be assessed by comparing one group's position to other groups in broader scenario |
| **Relative positions of stakeholders** | Drivers | Stakeholders who have high level of importance as well as high level of influence on public sector healthcare delivery system |
| | Supporters | Stakeholders who have high level of importance but low level of influence on urban healthcare delivery system |
| | Bystanders | Stakeholders who have low level of importance and low level of influence on urban healthcare delivery system |
| | Abstainers | Stakeholders who have no influence and no importance on urban healthcare delivery system |
| | Blockers | Stakeholders who have low level of importance but high level of influence on urban healthcare care delivery system. |

## Data analysis

A process flowchart for current decision-making practices will be prepared using the KII and organisational process reviews. A list of user needs will be made and shared at a stakeholder consultation meeting to identify the most important and feasible functions to be added to UHA.

For *qualitative data*, an outline plan for data analysis will be prepared in advance of research along with a priori codes. These codes, mostly focusing on user experience, will be derived from the WHO PATH toolkit.[19] The analysis will be open to emerging themes as well. All interviews will be recorded provided consent has been obtained, but with simultaneous note taking in case of equipment failure. Data transcription will occur immediately following each interview, followed by translation. Data familiarisation will involve reading transcripts repeatedly to surface emerging themes and identify any missed opportunities for further exploration. Transcripts will be coded using ATLAS-ti (V.7.5.7). A team approach to analysis will be employed to minimise individual biases. Inter-coder reliability will be checked. Group discussions of emerging themes and patterns in the data will be tested using data displays that allow more systematic pattern testing across respondents.

For *stakeholder analysis*, stakeholders' influence, importance and agreement will be explored applying standard techniques and measurement scales mentioned in table 3. A position diagram with level of agreement and level of influence will also be plotted to identify stakeholders who are already committed to work and help institutionalise UHA and those who need to be brought into agreement.

For the *policy engagement analysis*, KIIs with stakeholders will be examined to understand interactions between actors, content, context and processes with respect to ICT policy uptake, with a view to identifying entry points for policy advocacy and the sustainable incorporation of ICT in health systems. Interviews will be analysed using a priori codes drawn from the Policy Engagement Framework. Of additional interest in this analysis is understanding the mechanisms and processes of health policy development and how other policies may be important in efforts to integrate ICT into health systems. Based on these insights, a Health Policy for ICT process flowchart will be prepared. According to the Policy Engagement Framework, we will also seek information on how context is considered and dealt with when policies are formed and what processes need to be changed to more effectively integrate UHA into the system.[23]

For *quantitative data*, simple descriptive analysis will be performed to show user rates over time. The set of parameters to be analysed are as follows: number of users who accessed UHA, type of user, scope of UHA use and types of problems faced. For data analysis, software like MS Excel and STATA will be used as appropriate.

The *cost* of implementing UHA will be estimated using a direct approach. Average cost for each activity will be

calculated. All supply-side inputs will be identified, quantified and valued through record review and KIIs. Shared cost items, including salary, buildings, furniture, supervision, transportation and vehicles, will be identified through observation and interviews of relevant personnel. By considering the nature of inputs, these will be categorised into capital, recurrent as well as fixed and variable cost items. Shared costs will be apportioned by proportion of the time involvement of the relevant items (office rent, common vehicle) to different activities. Inputs will be identified using discussion with relevant personnel, observation and record review. If unavailable, market prices will be applied to estimate costs. The capital items will be annualised, and common costs will be apportioned as per requirement. Annual values of capital items will be estimated from their expected useful life years and annuitisation will be done using 3% discounting rate whenever applicable.[35] The study will allocate the cost for shared items (eg, office space, appliances) by using actual utilisation of items for this activity. Utilisation information for shared items will be collected from responsible project staff.

Finally, the cost for shared items will be estimated by multiplying percentage use information with the total cost of the items. Total cost will be calculated by summing up the start-up and implementation costs. Relative contribution of start-up and implementation cost will be calculated. The cost drivers in each activity will be identified considering the larger share of total cost.

## Process documentation
In addition to all these activities, the study investigators will systematically document implementation processes for policy uptake and institutionalisation of UHA, focusing particularly on contextual factors and their influence on implementation using a process documentation template. Process documentation of this 'pilot phase' of UHA institutionalisation will generate supporting knowledge to be applied the phases of scale-up and sustain, as specified in the WHO/PATH toolkit.

## Patient and public involvement
There will be no direct patient or public involvement in this implementation research. However, a technical advisory group (TAG) will be formulated for project governance, which will consist of representatives from government, development partners, NGOs, academicians and senior researchers and urban health actors. Regular meetings will be held with partners and staff for problem solving. At the end of the study, the TAG will comment on the study findings and contribute to the dissemination plan.

## Ethics and dissemination
The institutional review board of icddr,b is comprised of two committees: the Research Review Committee (RRC) and Ethical Review Committee (ERC). This study has received approval from both of the committees, which provided a thorough and critical review of the protocol's technical and ethical aspects. Participants will be asked for written consent prior interviewing and will remain anonymous and unidentifiable. Tape recorders will be used to record discussions but only after obtaining consent. All other forms of data will be kept in locked storage, or controlled access folders, allowing only investigators of the study and members of the ERC of icddr,b to access information, if needed.

Findings from this research will be disseminated at various levels to develop interest and support from a wide variety of audiences, that is, public, private, NGO, civil society and donors. In doing so, we hope to build a diverse constituency of individuals and organisations willing and able to translate evidence yielded by the study, into policy action.

### Local dissemination
Findings will be presented to relevant local administrators, development partners and NGOs, and other relevant parties (local health practitioners), researchers.

### National dissemination
A series of interactive workshops and briefing sessions with various stakeholders will be arranged to create linkages with national fora. The main aim will be to translate findings in a more visual and engaging formats, that is, research briefs and interactive project brochures, to reach a range of stakeholders.

### International dissemination
This will include publishing findings in peer-reviewed journals and presenting in scientific forums, conferences and symposiums, and linking with international learning platforms. The main objective is to contribute to global knowledge about context-specific strategies to incorporate ICTs into health systems, and challenges that must be anticipated for policy uptake necessary to introduce, scale-up and sustain MNCH-related ICTs in similar low-income and middle-income countries.

**Author affiliations**
[1]School of Population Health, University of New South Wales, Sydney, New South Wales, Australia
[2]Department of Family Medicine, Faculty of Medicine and Health Sciences, McGill University, Montreal, Quebec, Canada
[3]Universal Health Coverage Programme, Health Systems and Population Studies Division, International Centre for Diarrhoeal Disease Research Bangladesh, Dhaka, Dhaka District, Bangladesh

**Acknowledgements** This research study was funded by International Development Research Centre (IDRC). icddr,b acknowledges with gratitude the commitment of IDRC to its research efforts. icddr,b is also grateful to the governments of Bangladesh, Canada, Sweden and the UK for providing core/unrestricted support.

**Contributors** RI and AMA conceptualised the study. RI and AMA prepared the first draft of the manuscript. SMH, RA, DSB and SS revised the manuscript. RI and AMA reviewed critically for important intellectual content. SS revised the version submitted with inputs from all other co-authors.

**Funding** This work was supported by International Development Research Centre (IDRC), Canada (Grant Number: 108218-001).

**Competing interests** None declared.

**Patient consent for publication** Not required.

**Provenance and peer review** Not commissioned; externally peer reviewed.

**Data availability statement** Since this is a protocol paper, no data are available. Data will be handled according to the principles of the icddr,b policies and guidelines of the International Development Research Centre, Canada.

**ORCID iDs**
Rubana Islam http://orcid.org/0000-0002-9538-5390
Alayne Mary Adams http://orcid.org/0000-0002-0961-9825
Shaikh Mehdi Hasan http://orcid.org/0000-0002-3455-7072
Rushdia Ahmed http://orcid.org/0000-0001-6917-3266
Dipika Shankar Bhattacharyya http://orcid.org/0000-0003-0887-8937
Sohana Shafique http://orcid.org/0000-0002-5234-8522

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
