## [Reviewer comments · BMJ Open]

ARTICLE DETAILS

TITLE (PROVISIONAL)	Making Information and Communications Technologies (ICTs) work for health: protocol for a mixed-methods study exploring processes for institutionalizing geo-referenced health information systems to strengthen Maternal Neonatal and Child Health (MNCH) service planning, referral and oversight in urban Bangladesh
AUTHORS	Islam, Rubana; Adams, Alayne Mary; Hasan, Shaikh Mehdi; AHMED, RUSHDIA; Bhattacharyya, Dipika Shankar; Shafique, Sohana

VERSION 1 – REVIEW

REVIEWER	Gavin Yamey Duke Global Health Institute
REVIEW RETURNED	19-Sep-2019

GENERAL COMMENTS	Many thanks for asking me to review this protocol for a forthcoming study of a new information and communication tool that will be implemented in two city corporations and two municipalities in Bangladesh. I enjoyed reading this short protocol, which describes a potentially important and valuable "PPIR" (population, policy, and implementation research) study. While the protocol is generally well described and clear, there are some aspects that are hard to follow. I would recommend to the authors that they address two particular aspects: 1. It is difficult for readers to currently understand exactly what the actual intervention is. The ICT tool itself is not described in detail, and so it remains unclear what is being implemented. It seems to be some kind of atlas of facilities, but it is not clear what the status of this atlas is. Has it already been fully developed and is it now "ready for prime time"? Is this study going to be the first time it is being implemented in Bangladesh? Or is the tool itself going to be further developed during implementation (i.e. "learning by doing")? I think the paper would be much stronger if the tool were better described and if the authors could describe the tool in more detail and its implementation status.2. Right now, it is also unclear to readers how this tool could potentially improve health outcomes. The protocol will assess stakeholder perceptions of the tool, use of the tool, and costs of the tool. But what about assessment of whether using the tool improves public health outcomes? At the very least, what is the theory of change by which this atlas could impact such outcomes? The starting point for this study is the high mortality rate for
---

	children under five years (81 per 1000 live births) among urban slum residents in Bangladesh; reading the protocol, it remains unclear how the ICT tool could influence this rate. A few other smaller points:  - You note that "Nationally, mortality rate for children under five years of age is 65 per 1,000 live births while the rate is 81 per 1000 live births among urban slum residents in Bangladesh"; it would be good to also include the *rural* rate for comparison. - What are the sample sizes based on? (this is unclear) - Would be good to briefly say more about the "pluralistic health systems" that you mention and how these contribute to the high urban child mortality rate.
--	---

REVIEWER	Taufique Joarder FHI 360, Bangladesh Office
REVIEW RETURNED	19-Nov-2019

GENERAL COMMENTS	General comment: 1) This is an interesting protocol planned to study an under-researched area, i.e., the urban health system of Bangladesh. The methods are well described, but it mentions that it is implementation research (IR). But it did not mention what type of IR is this, e.g., pragmatic trial, effectiveness-implementation hybrid, quality improvement study, PAR, realist review, etc. Neither did it indicate any IR framework that the study based itself on. Therefore, I strongly recommend not to call it an IR, or integrate the essential elements of an IR into the protocol. Specific comments: 2) Table 1, row 1, column 5: What does the asterisk beside the 'Sample size' stand for. If it does not indicate anything, please remove it. 3) Table 1: User Need and User Experience: Authors have proposed 16 IDIs to determine user needs, and 16 IDIs on user experiences. It is not clear whether these respondents are separate respondents, i.e., 32 in total, or are they the same persons, i.e., 16 IDIs? If they are the same, then, please merge the relevant cells in a way that they do not give the impression that these are separate IDIs. If these are separate, what is the rationale for interviewing a separate set of the same type of respondents (Policymakers within the MoHFW, etc.)? 4) Page 19 of 35, line 17, Data collection: It is mentioned that the team consists of a qualitative & ethnography expert among others. What is the role of ethnography here? Just say, 'qualitative expert', if the role of the ethnographer is not justified. 5) Please avoid using bold (e.g., page 19 of 35, line 49) and italics fonts in paragraphs. The reviewer provided a marked copy with additional comments. Please contact the publisher for full details.
---

VERSION 1 – AUTHOR RESPONSE

Reviewer(s)' Comments:

Reviewer: 1

Reviewer Name: Gavin Yamey

Institution and Country: Duke Global Health Institute Please state any competing interests or state 'None declared': None declared

Please leave your comments for the authors below

Many thanks for asking me to review this protocol for a forthcoming study of a new information and communication tool that will be implemented in two city corporations and two municipalities in Bangladesh. I enjoyed reading this short protocol, which describes a potentially important and valuable "PPIR" (population, policy, and implementation research) study.

While the protocol is generally well described and clear, there are some aspects that are hard to follow. I would recommend to the authors that they address two particular aspects:

Comment1: It is difficult for readers to currently understand exactly what the actual intervention is. The ICT tool itself is not described in detail, and so it remains unclear what is being implemented. It seems to be some kind of atlas of facilities, but it is not clear what the status of this atlas is. Has it already been fully developed and is it now "ready for prime time"? Is this study going to be the first time it is being implemented in Bangladesh? Or is the tool itself going to be further developed during implementation (i.e. "learning by doing")? I think the paper would be much stronger if the tool were better described and if the authors could describe the tool in more detail and its implementation status.

Response: We have inserted a separate section to describe the functionalities of Urban Health Atlas. To respond to this comment the following texts have been added on page 9 and 10. We have also added in some additional information on how training around the UHA will be organized, and the kinds of training activities that are envisaged on page 17.

Comment 2: Right now, it is also unclear to readers how this tool could potentially improve health outcomes. The protocol will assess stakeholder perceptions of the tool, use of the tool, and costs of the tool. But what about assessment of whether using the tool improves public health outcomes? At the very least, what is the theory of change by which this atlas could impact such outcomes? The starting point for this study is the high mortality rate for children under five years (81 per 1000 live births) among urban slum residents in Bangladesh; reading the protocol, it remains unclear how the ICT tool could influence this rate.

Response: As per the suggestion of the reviewer we have revised the text and following sections have been added: on page 8 and page 29. Besides, a theory of change is provided in supplemental file 1.

A few other smaller points:

Comment: You note that "Nationally, mortality rate for children under five years of age is 65 per 1,000 live births while the rate is 81 per 1000 live births among urban slum residents in Bangladesh"; it would be good to also include the *rural* rate for comparison.

Response: Rural rate is added.

Comment: What are the sample sizes based on? (this is unclear)

Response: We have selected the sample size for qualitative component of the study based on available evidences. In order to clarify this, we have added the following lines at the end of "Sample size" section of the manuscript on page 17.

Comment: Would be good to briefly say more about the "pluralistic health systems" that you mention and how these contribute to the high urban child mortality rate.

Response: The pluralistic health systems, especially in the urban areas of Bangladesh, is associated with poor coordination among the authorities that results in poor planning and management with unclear roles and responsibilities for the various authorities and ultimately hampers the service coverage and human resource management issues. These issues have been already described in the text and to elucidate more the following line has been added with a reference on page 7.

Reviewer: 2

Reviewer Name: Taufique Joarder

Institution and Country: FHI 360, Bangladesh Office Please state any competing interests or state

'None declared': None declared.

Please leave your comments for the authors below General comment:

Comment: This is an interesting protocol planned to study an under-researched area, i.e., the urban health system of Bangladesh. The methods are well described, but it mentions that it is implementation research (IR). But it did not mention what type of IR is this, e.g., pragmatic trial, effectiveness-implementation hybrid, quality improvement study, PAR, realist review, etc. Neither did it indicate any IR framework that the study based itself on. Therefore, I strongly recommend not to call it an IR, or integrate the essential elements of an IR into the protocol.

Response: Thank you for pointing out where we could be more precise in our description of the study methods. This is an Implementation Research since this research is specifically aimed at looking at what factors influence the implementation of the UHA. We will explore the barriers and requirements for systematic implementing the UHA tool in the real world setting of the health managers. The primary audiences of this research are the managers and decision makers in urban health sector of Bangladesh.

We will explore the barriers and requirements for systematic implementing the UHA tool in the real world setting of the health managers.

We have now added more details to clarify how this study fits as an implementation research and what elements of an IR framework have been drawn upon in the development of the conceptual framework for this study. Following texts have been added on page 13 (methods section) and on page 14 (para 1).

Specific comments:

Comment: Table 1, row 1, column 5: What does the asterisk beside the 'Sample size' stand for. If it does not indicate anything, please remove it.

Response: We have removed the asterisk on page 15.

Comment: Table 1: User Need and User Experience: Authors have proposed 16 IDIs to determine user needs, and 16 IDIs on user experiences. It is not clear whether these respondents are separate respondents, i.e., 32 in total, or are they the same persons, i.e., 16 IDIs? If they are the same, then, please merge the relevant cells in a way that they do not give the impression that these are separate IDIs. If these are separate, what is the rationale for interviewing a separate set of the same type of respondents (Policymakers within the MoHFW, etc.)?

Response: The idea is we will conduct the user need interview to understand their preferences and then after the intervention we will conduct interviews with the same participants to get their perception and experiences about using the tool. Therefore the 16 IDIs will be of same participants. We accept this suggestion and formatting has been revised accordingly on page 16-17 (Table 1: Sampling strategy and sample type for each activity of the study).

Comment: Page 19 of 35, line 17, Data collection: It is mentioned that the team consists of a qualitative & ethnography expert among others. What is the role of ethnography here? Just say, 'qualitative expert', if the role of the ethnographer is not justified.

Response: We have taken out ethnography as per the suggestion of the reviewer on page 19.

Comment: Please avoid using bold (e.g., page 19 of 35, line 49) and italics fonts in paragraphs.

Response: We accept this suggestion and formatting has been revised accordingly on page 19.

VERSION 2 – REVIEW

REVIEWER	Taufique Joarder FHI 360, Bangladesh Office
REVIEW RETURNED	31-Mar-2020
GENERAL COMMENTS	The authors have adequately addressed the review comments. I approve the manuscript for publication from my end.